# Polymers for Osmotic Self-Inflating Expanders in Oral Surgical Procedures: A Comprehensive Review

**DOI:** 10.3390/polym17040441

**Published:** 2025-02-08

**Authors:** Alejandro Elizalde Hernandez, Natália Link Bahr, Matheus Coelho Blois, Carlos Enrique Cuevas-Suarez, Evandro Piva, Mateus Bertolini Fernandes dos Santos, Carla Lucia David Peña, Rafael Guerra Lund

**Affiliations:** 1Graduate Program in Dentistry, Pelotas Dental School, Federal University of Pelotas, Pelotas 96015-560, Brazil; chulets24@hotmail.com (A.E.H.); matheuscoelhoblois@hotmail.com (M.C.B.); carlosecsuarez@gmail.com (C.E.C.-S.); evpiva@gmail.com (E.P.); mateusbertolini@yahoo.com.br (M.B.F.d.S.); cldp58@gmail.com (C.L.D.P.); 2School of Dentistry, Federal University of Pelotas, Rua Gonçalves Chaves, 457, Room 503—Centro, Pelotas 96015-560, Brazil; nlinkbahr@gmail.com

**Keywords:** osmotic expanders, oral surgery, polymers, regenerative medicine, tissue expansion

## Abstract

Soft-tissue expansion is a critical challenge in regenerative oral surgeries. This scoping review maps the research on polymers used in osmotic self-inflating expanders, assessing their applications, characteristics, and potential in oral surgical procedures. The study analyzed 19 articles from the PubMed, Scopus, Web of Science, and Embase databases, primarily focusing on in vivo research (78.9%) investigating polymeric tissue expanders. The review examined polymer compositions, methodologies, and tissue responses across various animal models. Osmed^®^ hydrogel was the most studied material, with research exploring its expansion capabilities in rabbits, goats, pigs, rats, and beagle dogs. The findings showed diverse tissue expansion ranges and minimal inflammatory responses, indicating the potential for oral surgical applications. Despite promising results, gaps such as inconsistent expansion measurements and the lack of standardized protocols were identified. These findings highlight the need for further research to develop new polymer formulations and optimize device design to enhance safety, efficacy, and clinical predictability. This review provides a foundation for advancing polymeric tissue expander technologies, offering the potential for safer and more effective minimally invasive regenerative oral surgeries.

## 1. Introduction

The alveolar bone tissue depends on teeth and develops in parallel with tooth eruption. Its shape and size are determined by tooth roots. Tooth extraction causes inevitable bone resorption unless regenerative procedures are performed to prevent it [1]. Dental implants are often the first choice among various treatment options for replacing lost teeth. However, in cases of significant bone resorption, vertical and/or horizontal bone augmentation procedures are essential before implant placement to achieve the necessary dimensions of the alveolar ridge. Several surgical techniques, including block bone grafting, guided bone regeneration (GBR), vertical GBR, onlay and inlay grafting, and distraction osteogenesis, are employed in cases of severe bone resorption [2]. Despite their effectiveness, these techniques increase the risk of complications, mainly as to soft tissue dehiscence [3].

The precise management of soft tissue is essential for the success of guided bone regeneration and other tissue reconstruction techniques in the oral region [4]. Adequate soft tissue volume, elasticity, and vascularization are necessary to ensure optimal bone graft coverage, facilitating integration and preventing postoperative complications [5]. However, the quantity and quality of the available soft tissue is often insufficient, especially in areas with significant previous bone loss [6]. Surgical procedures for severe bone resorption require flap management to achieve soft tissue closure. However, this approach often promotes associated comorbidities, especially swelling and hematomas, which may compromise graft stability and increase the risk of complications, such as dehiscence and graft exposure [7].

These challenges have been addressed by various approaches, including the cortical tenting technique [8], the split-thickness flap design without vertical releasing incisions [9], the vestibular shifted flap design [10], conventional flap techniques, connective tissue grafts, and tissue expansion device incorporation [11]. Among these devices, osmotic self-inflating tissue expanders have emerged as an innovative and promising alternative. They are designed to gradually expand soft tissue in a less-invasive and controlled manner, utilizing the osmotic properties of the given polymers composing the device [12]. The action mechanism of these expanders is based on interstitial fluid absorption, which increases the device’s volume and, consequently, the expansion of the tissue covering it [13]. Furthermore, this process has the additional benefit of reducing the need for further surgical procedures, while simultaneously promoting a safer wound closure method and reducing the incidence of complications, such as retractions and graft loss [14].

The polymers employed in osmotic expanders are indispensable constituents, as their physicochemical attributes dictate the device’s expansion characteristics, biocompatibility, and mechanical resilience after degradation. Several in vitro [15] and in vivo [16] studies have investigated the properties of specific polymers, such as hydrogels, which have numerous desirable characteristics, including a high water-absorption capacity and biocompatibility with oral tissues. Furthermore, the ability to regulate the expansion profile, degradation rate, and pressure exerted on the tissue enables surgeons to tailor the tissue expansion approach to the requirements of each case. Extensive research has been conducted relative to Osmed^®^, a cross-linked hydrogel polymer network composed of methyl methacrylate and N-vinylpyrrolidone copolymers enveloped by a silicone shell [13].

Despite the potential of osmotic self-inflating expanders to promote controlled soft-tissue expansion, significant gaps remain in understanding their performance and clinical applicability [12]. The studies so far have concentrated on preclinical research in animal models [17], providing preliminary but unclear knowledge of biological responses and potential side effects of these devices in human clinical applications. Clinical studies have highlighted the effectiveness of tissue expanders combined with tunneled bone grafts for vertical augmentation. However, additional studies comparing this approach with alternatives that do not use tissue expanders are needed to elucidate the overall impact of tissue expansion. The current research focuses on the development of elastic hydrogels that retain the appropriate mechanical properties after swelling, maintain a controlled expansion rate, and prevent disintegration or tissue damage [18].

This scoping review aims to establish and critically evaluate the evidence on polymer use in osmotic self-inflating expanders for oral surgical procedures. It addresses the following key question: which polymers are used in the fabrication of osmotic self-inflating expanders? The review maps the existing evidence, identifying the utilized polymers and exploring their potential to improve these procedures’ predictability, safety, and minimally invasive nature.

## 2. Materials and Methods

### 2.1. Protocol and Registration

This review used the principles of the PRISMA-ScR (Preferred Reporting Items for Systematic Reviews and Meta-Analyses Extension for Scoping Reviews) protocol to ensure transparency and reproducibility in the selection and data extraction processes. However, specific adaptations were made in order to include a critical assessment of study quality and further synthesize the evidence, elements typical of a comprehensive review [19]. This study was registered in the Open Science Framework registries (Scoping Review Protocol: https://osf.io/vk8xy/ (accessed on 18 December 2024). The mapping followed parameters selected according to the PCC framework: (i) Population, tissue expansion; (ii) Concept, osmotic self-inflating expanders; (iii) Context, comparison of different polymers used in tissue expanders; (iv) Study Design, in vivo and laboratory studies. Although there was no classic PICO question, the review was guided by the following overarching research question: which polymer materials are used in tissue expanders used in oral surgical procedures?

### 2.2. Eligibility Criteria

The present work included studies evaluating swelling, expansion, and other properties of osmotic self-inflating tissue expanders, either in vitro or in vivo, and those assessing osmotic self-inflating expanders used alone or combined with other substances or biomaterials in oral surgical procedures. Four databases were searched: PubMed/MEDLINE, Scopus, Web of Science, and Embase. Only articles in English were included, there were no date restrictions, and the publication dates of the studies ranged up to December 2024. The exclusion criteria comprised studies not involving or reporting osmotic self-inflating expanders, those using conventional expanders with external ports for serial injections and manual inflation, and opinion articles. Studies on skin tissue expansion, systematic and scoping reviews, letters, articles not published in peer-reviewed journals, and conference abstracts were also excluded.

### 2.3. Search

The search strategy was based on MeSH terms from PubMed and adapted to other databases, using the keywords in Table 1. The grey literature was also considered, with the first 100 articles returned being retrieved from Google Scholar and OpenGrey.com.

### 2.4. Screening of Evidence Sources

Scientific and technological records were initially identified using EndNote X9 (Thomson Reuters, New York, NY, USA). Two trained and calibrated independent reviewers (AEH and CLDP) screened articles and patents for relevance and eligibility by analyzing titles and abstracts by using the Rayyan online software (Hamad Bin Khalifa University, Doha, Qatar). Records were classified into three categories: included, excluded, and uncertain. The same reviewers further assessed the full-text articles associated with included and uncertain records for eligibility. Discrepancies during title/abstract or full-text screening were resolved through discussion. A third reviewer was consulted in cases with a lack of consensus.

### 2.5. Data Charting Process

Data extraction used a pretested spreadsheet created in MS Excel (Microsoft Corporation, Redmond, Washington, DC, USA). This spreadsheet standardized data collection, and three reviewers (AEH, CLDP, and RGL) tested it to reach a consensus on collection variables and methods. One reviewer extracted the data, and the second verified the data’s accuracy and completeness.

### 2.6. Data Items

The primary outcome was the influence and/or use of osmotic self-inflating expanders before bone augmentation in dentistry. The following secondary data were collected from each study: author information (year, journal, and country), study characteristics (study design, sample size, control group, and inclusion criteria), and expansion polymers used (application methodology, use duration, and resulting tissue volume increase). The mapping considered the application of osmotic self-inflating expanders according to different dentistry areas.

## 3. Results

### 3.1. Selection of Evidence Sources

This review included 16 studies (Figure 1); Table 2 and Table 3 summarize the main findings. Most of the analyzed studies consisted exclusively of in vivo investigations (n = 11), followed by mixed studies that initially employed an in vitro design and subsequently conducted animal tests (n = 4) and purely in vitro studies (n = 1). The temporal analysis of the selected studies revealed a trend in scientific production over approximately 20 years, from 2009 to 2020, with fluctuating publication patterns characterized by periods of higher intensity interspersed with years of lower production (Figure 2).

### 3.2. Characteristics of Evidence Sources

Most reviewed studies primarily detailed the evaluated material, animal models, and measurement techniques. Therefore, the evaluation period of expanders was also considered.

#### 3.2.1. Material Type

In vivo analyses and mixed studies incorporating in vitro and in vivo designs represented the predominant research approach, comprising 78.9% of the reviewed articles. Notably, the in vitro and mixed studies, as well as most of the in vivo studies, used the Osmed^®^ self-inflating tissue expander, which was primarily applied for soft tissue expansion. This expander is a dehydrated hydrogel of a modified N-vinyl-2-pyrrolidone and methyl methacrylate copolymer [22,23,24,25,26]. Osmed^®^ emerged as the most frequently used material (Table 2 and Table 3), accounting for 57.8% of the studies. Its efficacy has been well-documented, demonstrating reliable expansion, biocompatibility, and a low incidence of postoperative complications [22,23,24,25,26]. Other materials, such as poly(ethylene glycol) (PEG) and poly(methyl methacrylate) (PMMA), were investigated in 26.3% of the studies, reflecting an ongoing search related to the optimization of expander materials. Additionally, 15.7% of the research examined combinations of Osmed^®^ with other polymers, which aimed to improve specific properties, such as deformation resistance and expansion rate flexibility.

#### 3.2.2. Selected Animal Models and Measurement Methods

These studies included various animal models, such as rats, rabbits, dogs, and pigs, introducing considerable heterogeneity among the results. Furthermore, data representation was based on linear dimensions and volumetric assessments. For instance, Abrahamsson (2009) [21] studied rabbits and reported a 5.5 mm expansion after one day of Osmed^®^ application, while Von See (2010) [23] observed no expansion in Lewis rats after 21 days of using Osmed^®^. Additional studies with rabbits demonstrated tissue expansion of 7.5 mm × 3 mm over 90 days [22,23,24]. Kaner (2015) [27] reported a volume increase of 141 mm^3^ in dogs after 14 days of Osmed^®^ application, further supporting its efficacy in inducing controlled tissue expansion. In vivo studies using animal models facilitate the extrapolation of results to clinical practice, providing a foundation for applying Osmed^®^ and other expanders in real-world clinical settings [22,23,24,25,26,27,28,29,30,31]. However, most studies (73.6%, n = 14) failed to report detailed expansion metrics, revealing a significant gap in methodological reporting and emphasizing the need for the standardization of data collection.

Some studies reporting expansion data can be categorized by measurement units: millimeters (mm), cubic millimeters (mm^3^), and cases without data reporting (NR) (Table 2 and Table 3). Notably, 73.6% of the studies did not report expansion data using standardized metrics, suggesting potential inconsistencies in experimental methodologies. Among the studies that reported the data, 15.7% used millimeters (mm), while 10.5% employed cubic millimeters (mm^3^), providing a more detailed volumetric analysis. The relatively limited use of cubic millimeters suggests that, although this technique offers a more comprehensive understanding of tissue expansion, it is employed less frequently in the research.

#### 3.2.3. Evaluation Period for the Expanders

Tissue expansion evaluation lasted from a minimum of 1 day [21] to a maximum of 1461 days [29]. However, this extended period was not associated with quantitative expansion measurements. Additionally, no complications regarding variations in placement duration or criteria for determining the evaluation period were reported. Notably, most studies documented an evaluation lasting no more than 90 days (Table 2).

## 4. Discussion

Osmotic self-inflating polymeric expanders are valuable in reconstructive surgery, particularly in the maxillofacial area, and have been proposed for oral implantology cases of severe bone resorption. They represent a promising advance in regenerative oral surgery, particularly for soft tissue preparation before bone grafting [27,31]. The findings suggest that these devices may significantly improve soft tissue quality, contribute to more efficient microcirculation, and facilitate bone regeneration [23].

The controlled and gradual expansion resulting from the use of these expanders allows progressive tissue adaptation, reducing the risk of trauma and increasing procedure predictability [16]. This is particularly relevant in oral surgery, in which mucosal fragility and critical anatomical structures require precision and control to ensure graft success [11,32]. Furthermore, flap advancement during guided bone regeneration reduces buccal height, potentially promoting cleansing difficulties, aesthetic issues, and lack of keratinized mucosa.

The first tissue expanders had limitations, such as the need for postoperative fillings, pressure spikes, and a filling valve far from the expander. Hence, a self-inflating tissue expander was developed. The self-inflating device comprises an osmotic hydrogel made from a methyl methacrylate (MMA) and vinyl pyrrolidone [33] copolymer.

Subsequently, new polymers and chemical designs for soft tissue expanders have been proposed based on the controlled expansion rate of the methacrylate hydrogel [15]. Additionally, the polymers used in these expanders, such as poly(ethylene glycol) (PEG) and poly(lactide-co-glycolide) (PLGA), have biocompatible properties that favor a minimal inflammatory response and the formation of thin and uniform fibrous capsules around the implanted material [33]. This is essential to prevent adverse reactions, such as excessive inflammation or fibrosis, which might compromise the function and aesthetics of the expanded tissue [34]. The formation of a controlled fibrous capsule also seems to help maintain the desired expansion volume, ensuring that the soft tissue remains prepared and sufficiently stable to receive the subsequent bone graft [35].

Despite these advances, current studies indicate the need for further improvements to optimize the effectiveness of polymeric tissue expanders in the clinical setting [36]. Several challenges remain, such as expansion rate adjustments to different tissue types and anatomical conditions and tissue response variability among individuals. The hyperperfusion rate in the first few weeks after placement, followed by stabilization at baseline, suggests the possibility of improving the initial phase of tissue adaptation to reduce discomfort or early complications such as edema [22].

Although current materials have demonstrated good resistance and expansion control, research on new polymers or hybrid compositions might provide an even more predictable response that is less susceptible to variations in temperature, humidity, and other environmental factors [37]. This aspect is noteworthy because these expanders are used in highly sensitive areas of the body which are subject to constant change, e.g., the oral cavity.

In the long term, developing devices that enable individual adjustments during expansion would be advantageous. This would allow clinicians to control the expansion rate and volume as needed [26]. This flexibility might help reduce dehiscence and optimize tissue adaptation, promoting a more harmonious bone graft integration with the prepared soft tissue [22].

Therefore, although osmotic self-inflating polymer expanders are innovative and promising for improving soft tissue preparation in regenerative oral surgery, additional studies are needed to refine their parameters and reduce the associated risks [12]. Human clinical trials and research on developing more conformable and durable polymers are essential to achieve broader and safer use of this technology. Ultimately, this advance might help improve bone grafts’ functional and aesthetic outcomes, providing more efficient regeneration and a more comfortable recovery for patients [38]. Conversely, hydrogel use for bone regeneration applications has also been investigated, showing promising results [39].

Considering the significant heterogeneity of tissue expansion outcomes and measurements, a meta-analysis was not feasible in this study. The differences in methodologies, animal models, and specific assessment areas contribute to data variability. These factors limit the direct comparability between the included studies and complicate the consolidation of results into a single quantitative analysis. It is worth noting that our study is a scoping review primarily aiming to map and synthesize the existing literature on the different polymers used in osmotic self-inflating expanders.

The variability in the current research requires more standardized and well-controlled studies. Future investigations should focus on standardizing methodologies.

## 5. Conclusions

This scoping review highlights the promising role of osmotic self-inflating expanders in addressing the challenges of soft tissue preparation in regenerative oral surgery. Through a comprehensive analysis of the current literature, the review highlights the pivotal contributions of polymer-based devices, mainly those using Osmed^®^ hydrogel, which has demonstrated favorable tissue expansion capabilities and minimal inflammatory responses across various animal models. Despite these advancements, significant gaps remain, including inconsistent methods of measuring expansion outcomes and a lack of standardized experimental protocols, hindering a broader clinical application of these technologies. These limitations may be addressed through future research focused on developing innovative polymer formulations and optimizing expander designs to enhance predictability, safety, and efficacy. Addressing these gaps may help in the evolution of the field towards establishing more effective and minimally invasive surgical strategies that improve patient outcomes in regenerative oral procedures. Potential future trends include innovative polymers as scaffolds in tissue engineering. This study is a foundation for further investigation and innovation in this critical research area.

## Figures and Tables

**Figure 1 polymers-17-00441-f001:**
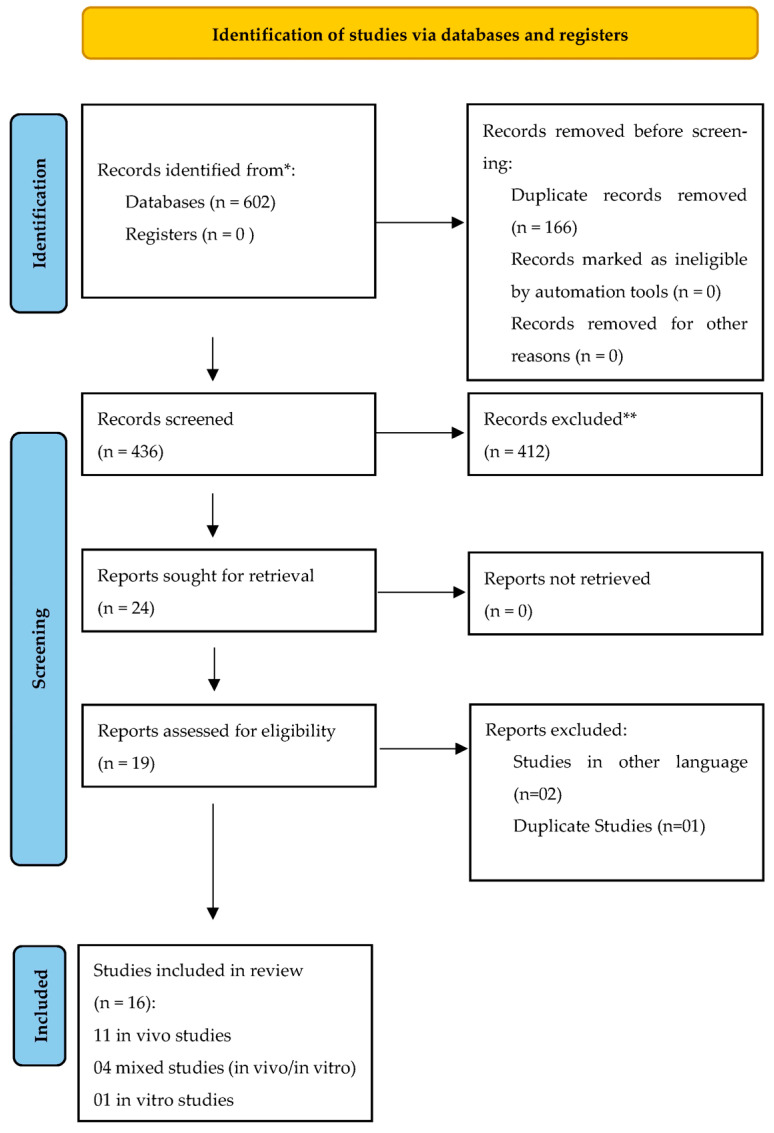
The following flowchart illustrates the selection of articles for inclusion in this study. * Consider, if feasible to do so, reporting the number of records identified from each database or register searched (rather than the total number across all databases/registers). ** If automation tools were used, indicate how many records were excluded by a human and how many were excluded by automation tools.

**Figure 2 polymers-17-00441-f002:**
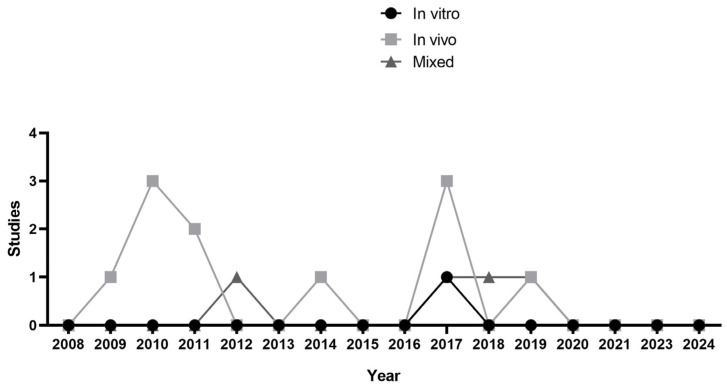
A review of in vivo and in vitro studies conducted over time.

**Table 1 polymers-17-00441-t001:** Search strategy.

Database	Keywords
PubMed/Medline	(((Tissue Expansion) OR (Expansion, Tissue) OR (Expansions, Tissue) OR (Tissue Expansions) OR (Controlled tissue expansion)) AND ((Tissue Expansion Devices) OR (Device, Tissue Expansion) OR (Devices, Tissue Expansion) OR (Tissue Expansion Device) OR (Tissue Expanders) OR (Expanders, Tissue) OR (polyhydroxyethylmethacrylate hydrogel) OR (Spheron 300) OR (Soflens) OR (2-hydroxyethyl methacrylate, N-vinyl pyrrolidone and 4-tertiary butyl-2-hydroxycyclohexyl methacrylate hydrogel) OR (polymacon) OR (Osmotic expanders) OR (Osmotic self-inflating expanders) OR (Self-Inflating Soft Tissue Expanders) OR (Osmotic expanders) OR (Self-filling osmotic tissue expanders))) AND ((Oral Surgical Procedures) OR (Surgical Procedures, Oral) OR (Procedures, Oral Surgical) OR (Surgical Procedure, Oral) OR (Oral Surgical Procedure) OR (Procedure, Oral Surgical) OR (Maxillofacial Procedures) OR (Maxillofacial Procedure) OR (Procedure, Maxillofacial) OR (Procedures, Maxillofacial) OR (Dentistry))
Scopus	(ALL ((“Tissue Expansion”) OR (“Expansion, Tissue”) OR (“Expansions, Tissue”) OR (“Tissue Expansions”) OR (“Controlled tissue expansion”) )) AND (ALL ((“Tissue Expansion Devices”) OR (“Device, Tissue Expansion”) OR (“Devices, Tissue Expansion”) OR (“Tissue Expansion Device”) OR (“Tissue Expanders”) OR (“Expanders, Tissue”) OR (“polyhydroxyethylmethacrylate hydrogel”) OR (spheron 300) OR (soflens) OR (“2-hydroxyethyl methacrylate, N-vinyl pyrrolidone and 4-tertiary butyl-2-hydroxycyclohexyl methacrylate hydrogel”) OR (polymacon) OR (“Osmotic expanders”) OR (“Osmotic self-inflating expanders”) OR (“Self-Inflating Soft Tissue Expanders”) OR (“Osmotic expanders”) OR (“Self-filling osmotic tissue expanders”) )) AND (ALL ((“Oral Surgical Procedures”) OR (“Surgical Procedures, Oral”) OR (“Procedures, Oral Surgical”) OR (“Surgical Procedure, Oral”) OR (“Oral Surgical Procedure”) OR (“Procedure, Oral Surgical”) OR (“Maxillofacial Procedures”) OR (“Maxillofacial Procedure”) OR (“Procedure, Maxillofacial”) OR (“Procedures, Maxillofacial”) OR (dentistry) ) )
Web of Science	(((Tissue Expansion) OR (Expansion, Tissue) OR (Expansions, Tissue) OR (Tissue Expansions) OR (Controlled tissue expansion)) AND ((Tissue Expansion Devices) OR (Device, Tissue Expansion) OR (Devices, Tissue Expansion) OR (Tissue Expansion Device) OR (Tissue Expanders) OR (Expanders, Tissue) OR (polyhydroxyethylmethacrylate hydrogel) OR (Spheron 300) OR (Soflens) OR (2-hydroxyethyl methacrylate, N-vinyl pyrrolidone and 4-tertiary butyl-2-hydroxycyclohexyl methacrylate hydrogel) OR (polymacon) OR (Osmotic expanders) OR (Osmotic self-inflating expanders) OR (Self-Inflating Soft Tissue Expanders) OR (Osmotic expanders) OR (Self-filling osmotic tissue expanders))) AND ((Oral Surgical Procedures) OR (Surgical Procedures, Oral) OR (Procedures, Oral Surgical) OR (Surgical Procedure, Oral) OR (Oral Surgical Procedure) OR (Procedure, Oral Surgical) OR (Maxillofacial Procedures) OR (Maxillofacial Procedure) OR (Procedure, Maxillofacial) OR (Procedures, Maxillofacial) OR (Dentistry))
EMBASE	((“Tissue Expansion”) OR (“Expansion, Tissue”) OR (“Expansions, Tissue”) OR (“Tissue Expansions”) OR (“Controlled tissue expansion”) )) AND (ALL ((“Tissue Expansion Devices”) OR (“Device, Tissue Expansion”) OR (“Devices, Tissue Expansion”) OR (“Tissue Expansion Device”) OR (“Tissue Expanders”) OR (“Expanders, Tissue”) OR (“polyhydroxyethylmethacrylate hydrogel”) OR (spheron 300) OR (soflens) OR (“2-hydroxyethyl methacrylate, N-vinyl pyrrolidone and 4-tertiary butyl-2-hydroxycyclohexyl methacrylate hydrogel”) OR (polymacon) OR (“Osmotic expanders”) OR (“Osmotic self-inflating expanders”) OR (“Self-Inflating Soft Tissue Expanders”) OR (“Osmotic expanders”) OR (“Self-filling osmotic tissue expanders”) )) AND (ALL ((“Oral Surgical Procedures”) OR (“Surgical Procedures, Oral”) OR (“Procedures, Oral Surgical”) OR (“Surgical Procedure, Oral”) OR (“Oral Surgical Procedure”) OR (“Procedure, Oral Surgical”) OR (“Maxillofacial Procedures”) OR (“Maxillofacial Procedure”) OR (“Procedure, Maxillofacial”) OR (“Procedures, Maxillofacial”) OR (dentistry) )

**Table 2 polymers-17-00441-t002:** Characteristics of the included in vivo (n = 11) and mixed (n = 4) articles.

Authors	Year	Journal	Country	Expansion Polymer	Study Type	Sample	n	Control	Inclusion Criteria	Experimental Animals	Expander Use Time	Increased Tissue Volume (mm or mm^3^)
Abrahamsson [20]	2009	*Scandinavian Journal of Plastic and Reconstructive surgery and Hand Surgery*	Sweden	Osmed	Animal study	Rabbits	8	NR	NR	Rabbits	1 day	5.5 mm (5.2–5.8 mm)
Von See [13]	2010	*International Journal of Oral & Maxillofacial Implants*	Germany	Osmed	Animal study	Lewis rats	48	Untreated group	NR	Rats	NR	NR
Abrahamsson [21]	2010	*Clinical Oral Implants Research*	Sweden	Osmed	Animal study	Rabbits	13	NR	Adult female Swedish lop rabbits	Rabbits	90 days	7.5 mm × 3 mm
Von See [22]	2010	*Clinical Oral Implants Research*	Germany	Osmed	Animal study	Lewis rats	16	NR	Isogenic male rats	Rats	21 days	NR
Uijlenbroek [23]	2011	*Clinical Oral Implants Research*	Netherlands	Osmed	Animal study	Goats	28	Untreated group	NR	Goats	40 days	NR
Abrahamsson [24]	2011	*Clinical Oral Implants Research*	Sweden	Osmed	Animal study	Rabbits	11	NR	Female Swedish lop rabbits	Rabbits	98 days	5.6 × 11 × 6 mm
Swan [25] Kaner [26]	2012	*Plastic and Reconstructive Surgery*	United Kingdom	Isotropic poly(methyl methacrylate-co-vinylpyrrolidone) hydrogels	Mixed study/Animal study	Pigs	6	Split-mouth design—untreated	Pigs	Pigs	4 days	Hydrogel X_2_ 540 mm^3^ Hydrogel X_6_ 900 mm^3^
Kaner [27]	2015	*Clinical Oral Implants Research*	Germany	Osmed	Animal study	Dogs	10	Granular biphasic calcium phosphate covered with polyethylene glycol membrane	Male beagle dogs	Dogs	14 days	NR
Barwinska [18] Jamadi [28]	2017	*Clinical Oral Implants Research*	Germany	Osmed	Animal study	Dogs	10	Split-mouth design—untreated	Jaws of healthy beagle dogs	Dogs	35 days	Experimental: 141 mm^3^ Control: 130 mm^3^
Yoo [17]	2017	*Plastic and Reconstructive Surgery–Global Open*	United States	Restiex HTE	Animal study	Dogs	9	Nearby mucous gingiva not surgically treated	NR	Dogs	42 days	21.5 mm^3^
Garner [16]	2017	*Macromolecular Bioscience*	United States	Poly(ethylene glycol) (PEG)	Animal study	Wistar rats	6	NR	NR	Rats	1461 days	NR
Hirb [29]	2018	*Journal of Periodontal & Implant Science*	Korea	Osstem	Mixed study/Animal study	Dogs	NR	NR	NR	Dogs	NR	NR
Ali Salim [30] Abrahamsson [20]	2019	*Journal of Periodontology*	United States	Cross-linked polymers of poly(ethylene glycol) (PEG) and poly(lactide-co-glycolide) (PLGA) connected by acrylate linkages	Animal study	Beagle dogs	9	NR	Beagle dogs	Dogs	42 days	Experimental: 8.13 mm Control: 6.44 mm
Von See [13]	2019	*Polymers*	Czech Republic	Poly(styrene-alt-maleic anhydride) covalently cross-linked with p-divinylbenzene	Mixed study/Animal study	Rats	9	NR	NR	Rats	29 days	12.7 mm^3^
Abrahamsson [21]	2020	*Materials*	Malaysia	Methyl methacrylate-N-vinylpyrrolidone copolymer (MMA-NVP)	Animal study	Dawley rats	7	Untreated group	Rats	Rats	28 days	NR

Abb: NR (not reported); Mixed (in vivo/in vitro studies).

**Table 3 polymers-17-00441-t003:** Characteristics of the included in vitro (n = 1) and mixed (n = 4) articles.

Authors	Year	Journal	Country	Expansion Polymers	Study Type	Control	Expander Use Time
Swan [25]	2012	*Plastic and Reconstructive Surgery*	United Kingdom	Poly(methyl methacrylate-co-vinylpyrrolidone)	Mixed/In vitro	Silicone and non-silicone cutlery	X_2_: 5 days X_6_: 22 days
Hrib [15]	2017	*Journal of Materials ScienceMaterials in Medicine*	Czech Republic	Poly(methyl methacrylate-co-vinylpyrrolidone)	In vitro	Polymer formulated with an anesthetic	40 days
Jamadi [28]	2017	*Macromolecular Bioscience*	United States	Poly(ethylene glycol) diacrylate (PEGDA)	Mixed/In vitro	NR	7 days
Yoo [17]	2018	*Journal of Periodontal & Implant Science*	Korea	Osstem	Mixed/In vitro	NR	NR
Hrib [29]	2019	*Polymers*	Czech Republic	Poly(styrene-alt-maleic anhydride)	Mixed/In vitro	NR	30 days

Abb: NR (not reported); Mixed (in vivo/in vitro study).

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
