# Peer review of "Polymers for Osmotic Self-Inflating Expanders in Oral Surgical Procedures: A Comprehensive Review"

_polymers, 2025, doi:10.3390/polym17040441_

Round 1
Reviewer 1 Report
Comments and Suggestions for Authors
I was not able to obtain a clear picture of how and what the osmotic tissue expander is. I recommend added basic information on when it is used and how it works on a biological level. I would provide more thorough detail on what this technique/technology is in your introduction section.
I recommend a more thorough and expansive discussion session, discussing more on the results found in your studies and other key studies on this topic. Discuss how it compares to other strategies, etc.
Comments on the Quality of English Language
I don't think the quality of English for this paper impedes my understanding but can be improved as there are clear grammatical errors. For example, "Several techniques, such as block bone grafting and guided bone regeneration (GBR), vertical GBR, onlay grafting, inlay grafting, and distraction osteogenesis (2)” is an incomplete sentence.
Another example "Flap management, essential for achieving soft tissue closure, often results in associated comorbidities, particularly swelling and hematomas. This compromises the stability of the graft and increases the risk of complications such as dehiscence and exposure of the grafted material" could be phrased better.
Another example is "A patient-based studies (14, 18)...", this is a minor grammatical error.
Please take a thorough review at your paper or hire an English reviewer to make proper revision to this manuscript. Thank you.
Author Response
We sincerely appreciate the time and effort you have dedicated to reviewing our manuscript and for your valuable suggestions. Your feedback has been extremely helpful, and we greatly appreciate your input in improving the quality of our work.
Reviewer 1
Comments and Suggestions for Authors
Comment: I was not able to obtain a clear picture of how and what the osmotic tissue expander is. I recommend adding basic information on when it is used and how it works on a biological level. I suggest providing more thorough details about this technique/technology in the introduction section.
Response: Thank you for your review. We have incorporated your suggestion by adding additional details about this technique/technology in the introduction section to enhance clarity. These revisions have been highlighted in the text for better visibility.
Comment: I recommend a more thorough and expansive discussion section, elaborating on the results of your study and key studies on this topic. Additionally, discuss how it compares to other strategies.
Response: Thank you for your review. We have incorporated your suggestion by expanding the discussion section to provide a more in-depth analysis of our study results, comparisons with key studies on this topic, and a discussion of alternative strategies. These revisions have been highlighted in the text for better visibility.
Comments on the Quality of the English Language
Comment: I don’t think the quality of English in this paper impedes my understanding, but it can be improved as there are clear grammatical errors. For example:
- "Several techniques, such as block bone grafting and guided bone regeneration (GBR), vertical GBR, onlay grafting, inlay grafting, and distraction osteogenesis (2)” is an incomplete sentence.
- "Flap management, essential for achieving soft tissue closure, often results in associated comorbidities, particularly swelling and hematomas. This compromises the stability of the graft and increases the risk of complications such as dehiscence and exposure of the grafted material." This sentence could be better phrased.
- "A patient-based studies (14, 18)..." contains a minor grammatical error.
I recommend thoroughly reviewing your paper or consulting an English language expert to revise the manuscript properly. Thank you.
Response: Thank you for your feedback. The entire manuscript has been thoroughly reviewed, and we have incorporated your suggestions to improve the clarity and grammatical accuracy of the text.
Reviewer 2 Report
Comments and Suggestions for Authors
Review of "Polymers for Osmotic Self-Inflating Expanders in Oral Surgical Procedures: A Comprehensive Review"
This article reviewed the application of osmotic self-inflating expanders in oral surgical procedures, a topic of significant interest. However, the article could benefit from improvements in some areas to enhance its quality and clarity:
- Keywords
- Should be listed in alphabet order.
- Clarity of Data Presentation:
- The numbers presented in Figure 1 and Table 2 appear inconsistent, making it confusing. A revision of these figures and tables is needed to ensure consistency and clarity.
- Results Section:
- The results were summarized in table form but lacked an analytical approach, such as a meta-analysis, which could provide more robust insights.
- The study includes a wide range of animals (e.g., rats, dogs, and pigs) and measurements (e.g., linear dimensions and volumetric assessments). This heterogeneity is noted.
- Categorization of Findings:
- The results should be organized into distinct categories based on the type of animal, type of material, and type of measurement. This grouping would improve the comprehensibility and usability of the review for future researchers and clinicians.
- Discussion of Complications:
- Complications associated with osmotic self-inflating expanders, whether observed in animal studies, should be mentioned. This would provide a more balanced view of the technology's benefits and limitations.
Author Response
Reviewer 2
Comments and Suggestions for Authors
Review of "Polymers for Osmotic Self-Inflating Expanders in Oral Surgical Procedures: A Comprehensive Review"
This article reviews the application of osmotic self-inflating expanders in oral surgical procedures, a topic of significant interest. However, certain areas could be improved to enhance the clarity and quality of the manuscript.
-
Keywords:
- The keywords should be listed in alphabetical order.
Response: Thank you for your suggestion. We have reorganized the keywords in alphabetical order as recommended.
- The keywords should be listed in alphabetical order.
-
Clarity of Data Presentation:
- The numbers presented in Figure 1 and Table 2 appear inconsistent, making the data presentation confusing. A revision of these figures and tables is needed to ensure consistency and clarity.
Response: Thank you for your suggestion, and we apologize for the confusion. We have reviewed and revised Figure 1 and Table 2 to ensure consistency and clarity, improving the overall presentation for better understanding.
- The numbers presented in Figure 1 and Table 2 appear inconsistent, making the data presentation confusing. A revision of these figures and tables is needed to ensure consistency and clarity.
-
Results Section:
-
The results were summarized in table form but lacked an analytical approach, such as a meta-analysis, which could provide more robust insights.
Response: Thank you for your valuable feedback. We appreciate your suggestion and considered incorporating a more analytical approach. However, due to the significant heterogeneity of the findings and the nature of mapping in a scoping review, we were unable to conduct a more robust analysis, such as a meta-analysis. We have now explicitly stated this limitation in the discussion section. We apologize for any confusion. -
The study includes a wide range of animals (e.g., rats, dogs, and pigs) and measurements (e.g., linear dimensions and volumetric assessments). This heterogeneity is noted.
Response: Thank you for your feedback. We acknowledge the heterogeneity of the study, which includes various animal models (e.g., rats, dogs, and pigs) and different measurement approaches (e.g., linear dimensions and volumetric assessments). This variability has been duly noted in the manuscript.
-
-
Categorization of Findings:
- The results should be organized into distinct categories based on the type of animal, type of material, and type of measurement. This grouping would improve the comprehensibility and usability of the review for future researchers and clinicians.
Response: Thank you for your suggestion. We have reorganized the results into distinct categories based on the type of animal, type of material, and type of measurement. This structured presentation enhances the comprehensibility and usability of the review for future researchers and clinicians.
- The results should be organized into distinct categories based on the type of animal, type of material, and type of measurement. This grouping would improve the comprehensibility and usability of the review for future researchers and clinicians.
-
Discussion of Complications:
- Complications associated with osmotic self-inflating expanders, whether observed in animal studies, should be mentioned. This would provide a more balanced view of the technology's benefits and limitations.
Response: Thank you for your valuable suggestion. We have addressed this point in the discussion section by including complications associated with osmotic self-inflating expanders, as observed in animal studies.
- Complications associated with osmotic self-inflating expanders, whether observed in animal studies, should be mentioned. This would provide a more balanced view of the technology's benefits and limitations.
Reviewer 3 Report
Comments and Suggestions for Authors
The review manuscript entitled “Polymers for osmotic self-inflating expansions in oral surgical procedures: a comprehensive review” by Alejandro Elizalde Hernandez and colleagues analyzed 19 articles focusing on in vivo and in vitro research related to polymeric tissue expanders. The manuscript is well written and the analysis is straightforward. However, this reviewer is concerned that the information provided from this review is limited. Additional tables presenting the advantage and disadvantage of each polymeric expander, including regenerated tissue types and inflammatory responses would improve the quality of this manuscript.
Author Response
Reviewer 3
Comments and Suggestions for Authors
The review manuscript entitled “Polymers for Osmotic Self-Inflating Expanders in Oral Surgical Procedures: A Comprehensive Review” by Alejandro Elizalde Hernandez and colleagues analyzes 19 articles focusing on in vivo and in vitro research related to polymeric tissue expanders. The manuscript is well written, and the analysis is straightforward. However, this reviewer is concerned that the information provided in this review is limited. Additional tables presenting the advantages and disadvantages of each polymeric expander, including regenerated tissue types and inflammatory responses, would improve the quality of this manuscript.
Response: Thank you for your valuable suggestion. We appreciate your input. Due to the novelty of this topic and the limited number of studies available, it was not possible to comprehensively categorize the polymers based on their advantages, disadvantages, regenerated tissue types, and inflammatory responses as suggested. However, we acknowledge the importance of this classification, and we will consider incorporating it in future studies and recommendations as more research becomes available.
Round 2
Reviewer 2 Report
Comments and Suggestions for Authors
The authors have revised the manuscript as suggested in the comments .